# An Edge-Based Architecture for Offloading Model Predictive Control for UAVs

**Achilleas Santi Seisa** , **Sumeet Gajanan Satpute** , **Björn Lindqvist** and **George Nikolakopoulos** *

Robotics and AI Team, Department of Computer, Electrical and Space Engineering,
Luleå University of Technology, 97187 Luleå, Sweden
* Correspondence: geonik@ltu.se

**Abstract:** Thanks to the development of 5G networks, edge computing has gained popularity in several areas of technology in which the needs for high computational power and low time delays are essential. These requirements are indispensable in the field of robotics, especially when we are thinking in terms of real-time autonomous missions in mobile robots. Edge computing will provide the necessary resources in terms of computation and storage, while 5G technologies will provide minimal latency. High computational capacity is crucial in autonomous missions, especially for cases in which we are using computationally demanding high-level algorithms. In the case of Unmanned Aerial Vehicles (UAVs), the onboard processors usually have limited computational capabilities; therefore, it is necessary to offload some of these tasks to the cloud or edge, depending on the time criticality of the application. Especially in the case of UAVs, the requirement to have large payloads to cover the computational needs conflicts with other payload requirements, reducing the overall flying time and hindering autonomous operations from a regulatory perspective. In this article, we propose an edge-based architecture for autonomous UAV missions in which we offload the high-level control task of the UAV's trajectory to the edge in order to take advantage of the available resources and push the Model Predictive Controller (MPC) to its limits. Additionally, we use Kubernetes to orchestrate our application, which runs on the edge and presents multiple experimental results that prove the efficacy of the proposed novel scheme.

**Keywords:** edge computing; UAV; model predictive control; Kubernetes

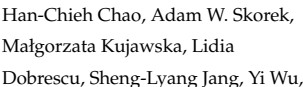



## 1. Introduction

Today, the need for autonomous solutions in robotics is rapidly increasing, while Unmanned Aerial Vehicles (UAVs) are used more and more every day for scenarios that range from inspection, maintenance, mapping and exploration to surveillance, photography and transportation uses. Some of these missions are quite complex, especially when the environments in which the UAV must operate are harsh and unknown, a great amount of data is generated from sensors and computationally heavy algorithms are required to ensure the desired levels of autonomy. For some of these tasks, the data processing and overall algorithmic operation must happen in real-time and in a bounded time delays perspective. In this article, the goal is to control the trajectory of a UAV by offloading the control architecture to an edge server. For this reason, we used a Model Predictive Controller (MPC), which is extremely effective but can be computationally heavy. The computational demands of the controller are due to the MPC optimization method, which can have a high complexity. Additionally, the chosen MPC horizon can be long and, consequently, require a significant amount of computational resources for performing the needed optimizations in fixed and bounded time instances. Taking into consideration the high computational demands of the MPC, we propose a novel edge-based architecture to offload the MPC to the edge. At the same time, we propose the utilization of Kubernetes architecture for the overall orchestration. The edge server will give us the opportunity to

experiment with various MPC horizons and rates in several use cases, while at the same time the system will not suffer from significant time delays. In contrast, Kubernetes will automate the deployments, scaling and managing of the control application, which runs inside a docker container.

### 1.1. Background and Motivation

Edge computing has piqued the interest of many researchers and engineers. The capabilities can be significant in many cases, such as autonomous vehicles, traffic management, remote monitoring, smart grid, predictive maintenance, patient monitoring, content delivery, smart homes and even more areas [1]. In robotics, edge computing is gaining remarkable attention, and researchers are trying to integrate edge capabilities into their applications. In [2], a system architecture for offloading SLAM tasks to the edge is presented. The system consists of four layers: the robot and edge layers for data processing and analysis, a fog layer for distributed storage and a cloud layer for monitoring and general mission control. In this case, the edge is used for real-time decisions thanks to minimal latency times. In [3], a system to teleoperate dynamic self-balancing robots that can detect and pick up objects through the edge (dynamic self-balancing navigation controller) and cloud (assisted teleoperation and visual recognition) is demonstrated, while in [4], researchers approached the task of object recognition and grasp planning by a mobile robot, where deep robot learning distributes computation, storage and networking resources between the cloud and the edge. In [5], a fog robotics architecture is proposed, in which the response rate of the robots is validated, and the whole system is examined in terms of latency. A search planner algorithm using deep learning is designed at the edge for UAVs in [6].

In this work, we focus on proposing a novel architecture for offloading the high-level and computationally heavy motion controller to the edge in order to control the trajectory of a UAV. As in our case, MPC has been, in general, proposed for offloading to the edge mainly for process control-oriented approaches, as is depicted in the following articles, in which related architectures have been evaluated in terms of their latencies and uncertainties. In comparison to our work and the proposed framework, we are ahead of the state of the art, and we propose a novel control and robotics-oriented architecture with trending technologies, such as Kubernetes. In [7], the overview of an edge/cloud architecture is described, and an example is presented to evaluate a remote MPC to control a ball and beam system, while an architecture with two offloaded MPCs (on the edge and the cloud) is described in [8,9]. In [10,11], the system is composed of multiple controllers again, but, this time, there is an LQR locally, which does not require many computational resources, and a much more demanding MPC is placed on the edge. Researchers in [12] evaluated the strategy of a system by changing the values of the MPC horizon. In [13], the authors proposed offloading the computationally heavy MPC to multiple ground-based computational units (CUs). This work is focused on an architecture in which multiple ground CUs carry on the task of controlling multiple UAVs, even when the CUs are less than the UAVs. This architecture is characterized by the event-triggered distributed MPC and is evaluated through a simulation setup.

Sometimes, the edge is used for offloading complex procedures in industrial environments. In [14], applications are deployed in docker containers, which run on a mobile edge server. The containers contain remote controllers for controlling the two robot arms in an industrial environment. The goal of the robotic arms is to complete cooperation tasks while receiving commands from the edge. In [15], researchers were concerned about safety issues; this is why they implemented a switching multi-tier control, a demanding controller with sophisticated algorithms, while a safety controller runs locally. The switcher is responsible for choosing between the two controllers, or, in other words, for switching from the edge to the safety controller in case of an emergency. A switching mechanism is also presented in [16], where the offloading decision mechanism will decide between a local or remote (edge-based) operation of path planners and estimators for a mobile robot. Advanced

algorithms, which are developed in containerized form and offloaded to the edge, are used when the switching mechanism favors the remote operation.

In the proposed novel architecture, we introduce a Kubernetes component-oriented architecture and novelly deploy our application, specifically oriented for the case of robots on a Kubernetes cluster. Kubernetes are widely used in cloud and edge computing solutions, but their potential in the robotics community has not been explored until now. In [17], an optimization method is presented for distributing containers, which applies AI and feature extraction methods to big data collected from various sensors from smart homes, cities, construction sites and robots. The containers are distributed to the cloud, edge and fog, based on a decision method and depending on the application requirements. The decision-making method is based on stochastic processes, while an architecture that automates the whole process is designed and implemented using Kubernetes' orchestration. In [18], an edge architecture and an open-source network are introduced for distributed edge and cloud resources. In this work, Kubernetes is used for orchestrating a virtual cluster comprising Virtual Machines (VMs). Container orchestration is also the subject of [19], in which researchers focused on providing low-latency edge services for robotic applications while containers interact with each other through plugins implemented based on Container Network Interface (CNI). Finally, similar frameworks with a different architecture are presented in [20]. In this case, an architecture using Docker, Kubernetes and a Robotic Operating System (ROS) is introduced in which the robotic applications are organized in a ROS framework and deployed into cloud clusters. The system's architecture was evaluated through the experimental results of a mobile robot interacting with an industrial agile production chain; however, this approach did not focus on the control-oriented needs or the offloading key performance components on the edge, e.g., the key high-level motion planning framework.

Though VMs have been used widely for offloading robotic applications to the edge, the benefits of a containerized application and Kubernetes constitute a preferred infrastructure for our architecture. Since containers only emulate software components, such as system libraries, external software packages and other operating system level applications [21], they are more lightweight and easier to iterate, whereas VMs need much more time to be deployed because they emulate an entire machine down to the hardware layers. Our containerized MPC application can be deployed in any cluster rapidly and easily. Furthermore, some robust pre-made container images, such as the ROS image used in this work as an entry point, are provided. Finally, multiple software packages can be deployed in multiple images to produce a novel and more complete application (we used two different images for our application). To fully benefit from the advantages of containerized applications, container orchestrators, such as Kubernetes, are needed. Kubernetes provides reliability, scalability, robustness and security, which are some essential requirements in robotic applications [22]. Kubernetes performs multiple essential tasks, such as reducing the network overhead, increasing the resource usage efficiency, designating hardware resources for your specific configuration and monitoring nodes and components, which leads to the smooth execution of the controller on edge. These procedures in a VM environment should be done manually by the user.

### 1.2. Contributions

The main goal and contribution of this article is to take advantage of the edge capabilities and establish an edge-based architecture for offloading the high-level MPC-based motion controller over the edge using a Kubernetes orchestration for enabling autonomous UAV missions. MPC is a computationally demanding controller; therefore, by offloading the controller to the edge, we will have the opportunity to run experiments with various MPC horizons and rates while satisfying the real-time characteristics of the application and ensuring round-trip bounded time delays. Without the existence of the link to the edge, we would not be able to run these experiments efficiently because the UAV's computational power cannot afford to run these processes without violating the real-time bounds. In this

article, we propose a novel edge-based architecture for offloading the MPC to the edge and evaluate the architecture's capabilities in terms of latency and computational power. This architecture can also be used in the future for offloading additional more complex and computationally heavier processes to the edge; thus, we can move towards fully edge-oriented ubiquitous autonomous solutions. In the novel established architecture, as will be presented, we concentrate on the edge layer in which the applications are developed with Kubernetes orchestration, and the MPC, optimizer and ROS master are deployed inside PODs (Figure 1).

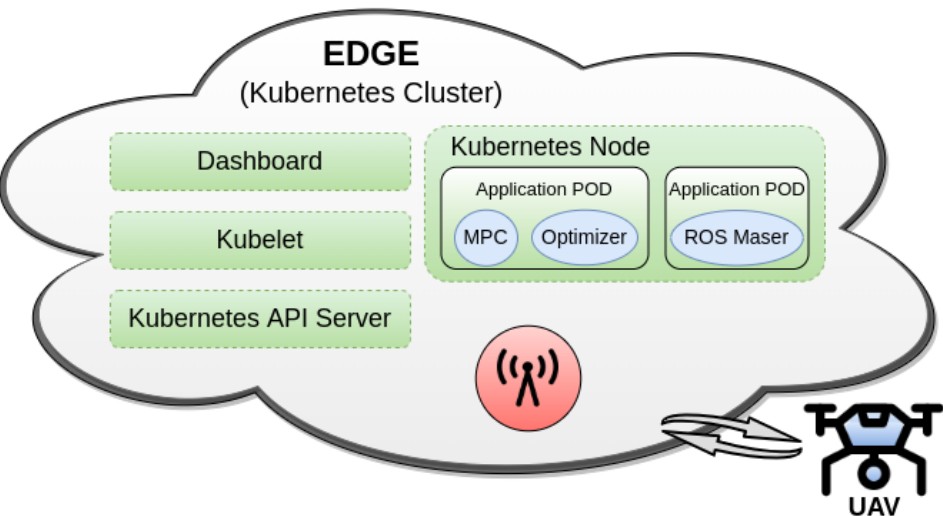

**Figure 1.** Kubernetes-based edge architecture for controlling the trajectory of one UAV via an MPC that runs on the edge.

### 1.3. Outline

The rest of the article is structured as follows: In Section 2, the model predictive control scheme is presented, and, in Section 3, we describe the novel system's architecture with Kubernetes orchestration oriented for robotic applications. Most of the edge providers offer Kubernetes solutions because of the advantages they provide. On the other hand, there are some challenges that we might face when using Kubernetes, which will be addressed. Additionally the parameters of the controller are presented. In the sequel, the architecture and the software used are presented in Section 4, in which we also evaluate the usage of the edge/Kubernetes cluster in matters of latency and computation power. Future research directions and additional tasks that can be assigned to the edge are mentioned in Section 5, including the conclusions of the article.

## 2. Model Predictive Control

### 2.1. UAV Kinematics

Model Predictive Control has been widely used in both research and industrial environments. In this article, MPC is implemented for following the desired trajectory of a UAV that is modeled as a six-degrees-of-freedom robot with a fixed body frame, and its kinematic model can be described by Equation (1) in the body frame as [23]:

$$\dot{p}(t) = v_z(t)$$

$$\dot{v}(t) = R_{x,y}(\theta, \phi) \begin{bmatrix} 0 \\ 0 \\ T \end{bmatrix} + \begin{bmatrix} 0 \\ 0 \\ -g \end{bmatrix} - \begin{bmatrix} A_x & 0 & 0 \\ 0 & A_y & 0 \\ 0 & 0 & A_z \end{bmatrix} u(t) \qquad (1)$$

$$\dot{\phi}(t) = \frac{1}{\tau_\phi}(K_\phi \phi_d(t) - \phi(t))$$

$$\dot{\theta}(t) = \frac{1}{\tau_\theta}(K_\theta \theta_d(t) - \theta(t))$$

In Equation (1), $p = [p_x, p_y, p_z]^T$ is the position, and $v = [v_x, v_y, v_z]^T$ is the linear velocity referenced in the global frame. $R(\phi(t), \theta(t)) \in SO(3)$ is the rotation matrix that describes the attitude in Euler form, while $\phi$ and $\theta \in [-\pi, \pi]$ are the roll and pitch angles along the $x^W$ and $y^W$ axes, respectively. Furthermore, $\phi_d$ and $\theta_d \in R$ and $T \geq 0$ are the desired input values to the system for the roll, pitch and total thrust. The above model assumes that the acceleration is only dependent on the magnitude and angle of the thrust vector, produced by the motors, as well as the linear damping terms $A_x, A_y, A_z \in R$ and the gravity of earth $g$. The attitude terms are modeled as a first-order system between the attitude (roll/pitch) and the desired $\phi_d$ and $\theta_d \in R$, with gains $K_\phi$ and $K_\theta \in R$ and time constants $\tau_\phi$ and $\tau_\theta \in R$. It is assumed as well that the UAV is equipped with a lower-level attitude controller that takes thrust, roll and pitch commands and provides motor commands for the UAV.

*2.2. Cost Function*

For the cost function, the UAV's state vector is represented as $x = [p, v, \phi, \theta]^T$, and the control input as $u = [T, \phi_d, \theta_d]^T$. The system has a sampling time of $\delta_t \in \mathbb{Z}^+$ using a forward Euler method for each time instance $(k+1|k)$. This discrete model is used as the prediction model of the MPC. The prediction considers a specified number of steps into the future, called the prediction horizon, which is denoted as $N$. An optimizer is tasked with finding an optimal set of control actions, defined by the minimum of this cost function, by associating a cost to a configuration of states and inputs at the current time and in the prediction. The predicted state at time step $k+j$, produced at the time step $k$, is represented as $x_{k+j|k}$. The corresponding control actions are represented as $u_{k+j|k}$. Additionally, $x_k$ and $u_k$ represent the fully predicted states and the corresponding control inputs along the prediction horizon. The objective of the controller is to navigate to the desired position and deliver smooth control inputs, and the cost function is presented in Equation (2).

$$J = \sum_{j=1}^{N} \underbrace{(x_d - x_{k+j|k})^T Q_x (x_d - x_{k+j|k})}_{state \quad cost}$$

$$+ \underbrace{(u_d - u_{k+j|k})^T Q_u (u_d - u_{k+j|k})}_{input \quad cost} \qquad (2)$$

$$+ \underbrace{(u_{k+j|k} - u_{k+j-1|k})^T Q_{\delta u} (u_{k+j|k} - u_{k+j-1|k})}_{control \quad actions \quad smoothness \quad cost}$$

where $Q_x \in \mathbb{R}^{8x8}$ is the matrix for the state weights, $Q_u \in \mathbb{R}^{3x3}$ is the matrix for the input weights and $Q_{\delta u} \in \mathbb{R}^{3x3}$ is the matrix for the input rate weights. In (2), the first term describes the state error cost, which is the cost penalty associated with the deviation from a certain desired state $x_d$. The second term describes the input cost that penalizes a deviation from the steady-state input $u_d = [g, 0, 0]$ and represents the inputs that describe hovering. The final term is added to guarantee that the control actions are smooth, which is achieved by comparing the input at $(k+j-1|k)$ with the input at $(k+j|k)$ and penalizes

the changing of the input from one time step to the next one, with $N \in N^+$ to denote the control horizon of the MPC.

### 2.3. Constrain Formulation

Except for the cost function, we defined some constraints regarding the control input rate and the control inputs. To restrict the aggressive and oscillatory behavior of the controller, we impose these constraints to guarantee successive differences in the control actions and bounds on their changes. Thus, we set a bound on the magnitude of the change in control inputs $\phi_{ref}$ and $\theta_{ref}$ described by the following equation:

$$
[\phi_{ref,k+j-1|k} - \phi_{ref,k+j|k} - \delta\phi_{max}]_+ = 0
$$
$$
[\phi_{ref,k+j|k} - \phi_{ref,k+j-1|k} - \delta\phi_{max}]_+ = 0. \tag{3}
$$

The same constraints are imposed for $\theta$ with $\delta\phi_{max}$ and $\delta\theta_{max}$ as the maximum change in input.

Moreover, we applied constraints on the control inputs to stabilize the attitude. This constraint is applied by defining bounds on the thrust input given as:

$$
u_{min} \leq u_{k+j|k} \leq u_{max}. \tag{4}
$$

In this work, we focus on evaluating the behavior of an edge-implemented overall MPC architecture by selecting various values for the horizon and the execution rate. To that end, we also evaluate the behavior of the system with several experiments, as presented in Section 4. Furthermore, in Sections 2.3.1 and 2.3.2, we present the reason for running multiple experiments with different values for these parameters.

### 2.3.1. MPC Prediction Horizon

One important parameter of the MPC is the prediction horizon because it sets a finite time-horizon on which the current timeslot will be optimized, while future timeslots will be taken into account in a repetitive approach. The MPC horizon provides the ability to anticipate future events and can take control actions accordingly, which can be extremely handy in dynamic environments, as stated in [24]. A longer horizon might mean more accurate predictions, but these advantages do not come without a cost. Since the optimization method is based on the prediction horizon, the longer the horizon is, the more computational resources are needed. That is one of the reasons that we propose and evaluate an edge-based architecture in Section 4.

### 2.3.2. MPC Rate

The second parameter with which we decided to experiment is the MPC rate. This parameter is partially responsible for the execution time of the MPC, since running the MPC at a high frequency means that control commands generate much faster. This is especially crucial in our case because the generated commands have to travel from the edge to the UAV. In the proposed novel architecture, we have two types of time delays. The first is based on the execution time of the MPC $d_2$, and the second is the time the signal requires to travel from the edge to the UAV and vice versa $d_1$, as shown in Figure 2. These two time delays comprise the round-trip time delay. In the presence of the travel time and execution time delays, there is a mismatch between the time stamp of the CF states, for which the control command was calculated, and the present CF state, for which the control command is executed. This mismatch can cause the UAV to fly to a random position, instead of the desired one, if the delays are relatively long. In robotic-oriented control frameworks, it is paramount to have a low round-trip time delay so that the UAV is able to react to commands faster. By choosing a fast MPC (with an increased MPC rate), we can reduce the round-trip time by reducing the execution time.

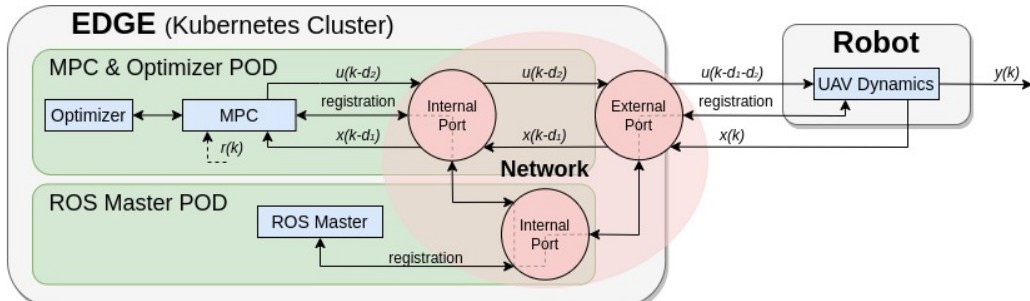

**Figure 2.** Block diagram of the edge architecture for the UAV-MPC system.

### 3. Edge Architecture

The application PODs contain Docker containers that run the MPC and ROS master on the edge, while in this case, Kubernetes is used for the orchestration. A container is a unit of software that runs code and all the dependencies so that the application deployed on the container will run quickly and reliably from any computer. A POD is a group of containers that shares storage and network resources, and Kubernetes is an orchestration system for automatic application deployment, scaling and management [25]. In our work, we used two Ubuntu container images with a ROS for deploying the MPC and ROS master on the edge, as shown in Figure 1. The two images are similar, but, in one, we deployed the MPC and the optimization method, and, in the other one, we deployed the ROS master.

In the presented edge architecture, the MPC commands are sent from the edge to a UAV with low or no computational capabilities, as shown in Figure 1. This figure also represents the basic structure of the system. In the proposed implementation, we utilized the TCP protocol and the host network to map the listening port of the edge to the port of each POD, as well as for overall communication between the edge and the local computer. Furthermore, we ran the ROS master, the MPC and the optimizer ROS nodes on the edge. In more detail, the ROS master runs on an application POD, while the MPC and the optimizer run on another application POD. The architecture and the block diagram of the system are shown in Figure 2. The MPC and the optimizer were deployed in the same POD. The MPC ROS node subscribes to the odometry topic and the reference topic in order to receive the states necessary for controlling the UAV and publishes the commands to the command velocity topic in order to send the commands back to the UAV. On the other side, the UAV subscribes to the command velocity topic to receive the commands and publishes its position to the odometry topic. Both the UAV and the MPC node are registered to the ROS master in order to establish communication between them.

In Figure 2, we can observe the parameters of the system. In order to control the UAV, we use the reference signal that describes the desired pose and the states signal, which describes the real pose of the UAV. The reference input signal for the desired trajectory is depicted as $r(k)$, and the states signal generated by the UAV dynamics are depicted as $x(k)$. These states are the linear and angular positions and velocities of the UAV. For closing the overall control loop, we should feed these states signals back to the MPC. As the MPC is running on the edge, we have to indicate the latency times and, as such, $d_1$ represents the time delay between the time for which the UAV dynamics generated the states signal on the local computer to the time in which the signal gets to the MPC on the edge. This is why the states signal that arrives to the MPC is represented as $x(k - d_1)$. We depict the command signal generated by the MPC as $u(k - d_2)$. The value $d_2$ denotes the MPC's execution time depending on the MPC rate and the computational process. Again, since the command signal has to travel from the edge to the local computer, where the UAV model is implemented, the command signal arriving to the UAV is $u(k - d_1 - d_2)$. This command signal also represents the necessary thrust for each one of the rotors for the UAV to follow the desired trajectory. Finally, $y(k)$ is the output of the system, which describes the $x, y, z, yaw$ values of the real pose of the UAV.

## 4. Experimental Results and Evaluation

To evaluate the architecture of our system, we used a Crazyflie (CF) [26], shown in Figure 3, which is a small UAV without computational capabilities. In order to control the CF, we ran the MPC on the edge and set various different MPC horizons and rates. The behavior of the CF is depicted in the following figures. To extract these figures, we used the Matlab software MathWorks. The edge layer used for these experiments consisted of a Kubernetes cluster located in Luleå, Sweden, within the same network in which we were operating the CF, and provided computational resources with minimal delays.

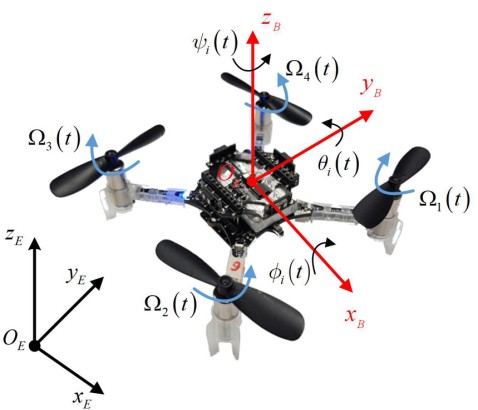

**Figure 3.** The UAV Crazyflie used for the experimental evaluation. CF is a resource-constrained UAV that is not capable of running the MPC on board.

Based on the performed experiments, we recorded ROSbags of the UAV while flying and present the corresponding results below. In Figure 4, a three-dimensional view of the trajectory of the CF is depicted. Figures 5–9 show the responses for the three frame axes ($X$, $Y$, $Z$) of the CF, while flying in a circular trajectory, for several different horizons, running at five different MPC rates. Moreover, the error between the reference trajectory and the real position values for each frame axis is presented in Figure 10. Finally, the round-trip time is shown in Figure 11 for different MPC rates, while the prediction horizon is set at 100 steps.

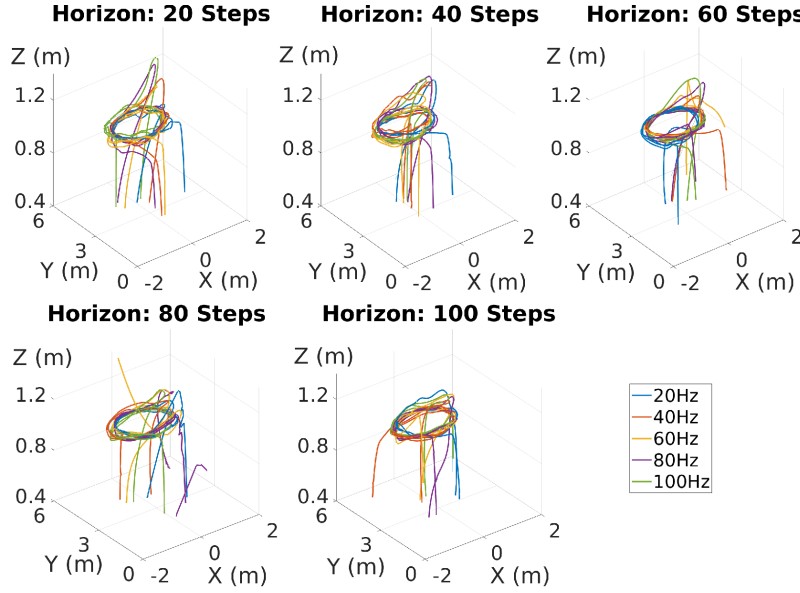

**Figure 4.** The 3D views of the circular trajectories for various MPC rates (from 20 Hz to 100 Hz) and MPC horizons of 20, 40, 60, 80 and 100 steps.

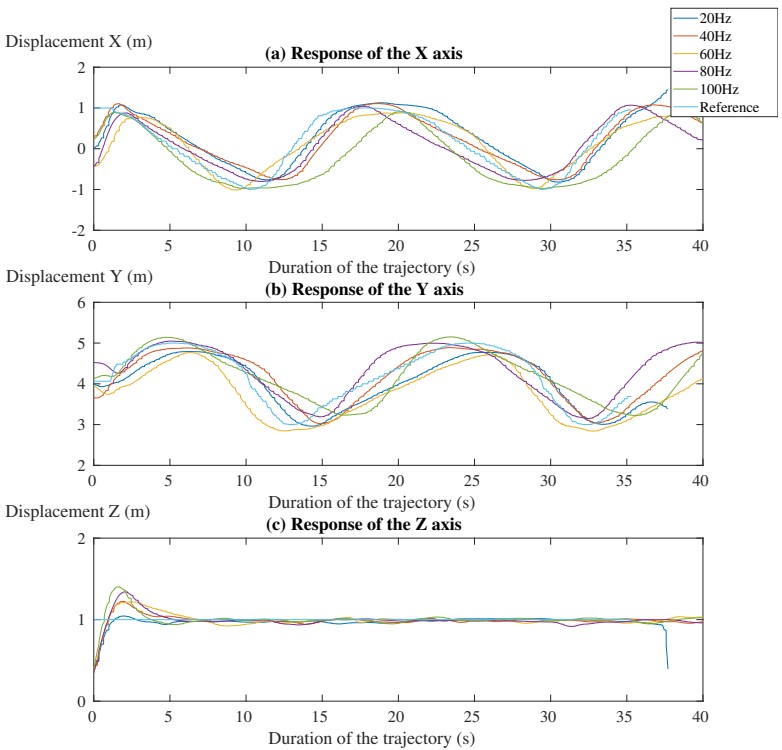

**Figure 5.** Response for a circular trajectory for various MPC rates and an MPC horizon of 20 steps.

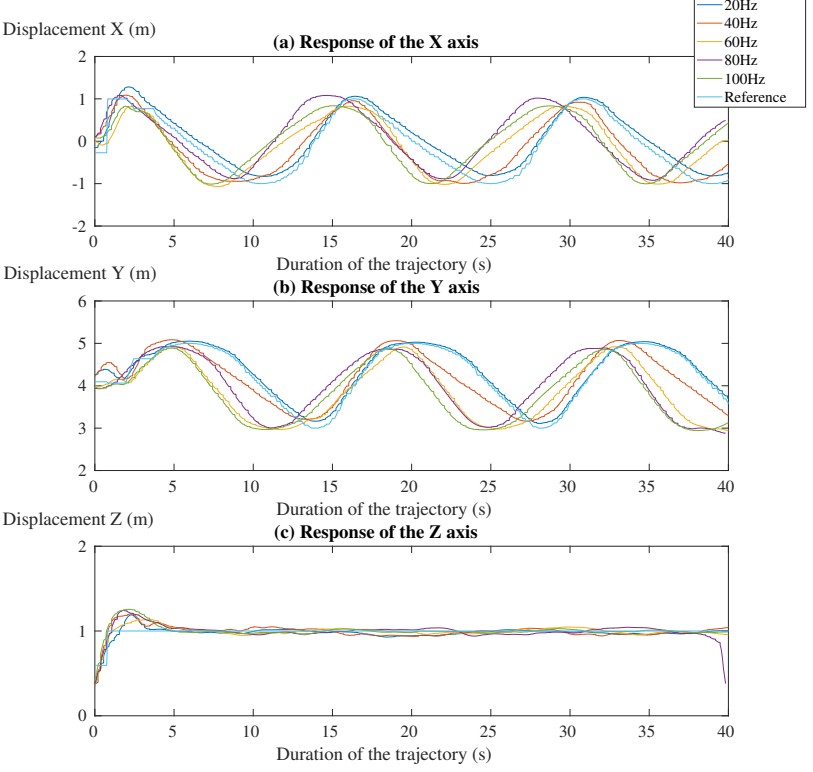

**Figure 6.** Response for a circular trajectory for various MPC rates and an MPC horizon of 40 steps.

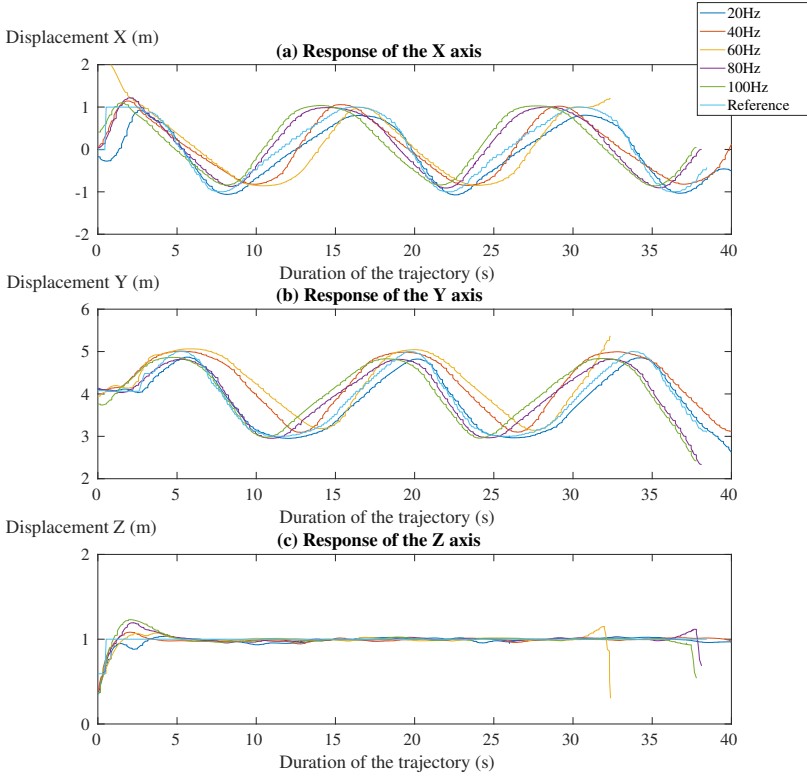

**Figure 7.** Response for a circular trajectory for various MPC rates and an MPC horizon of 60 steps.

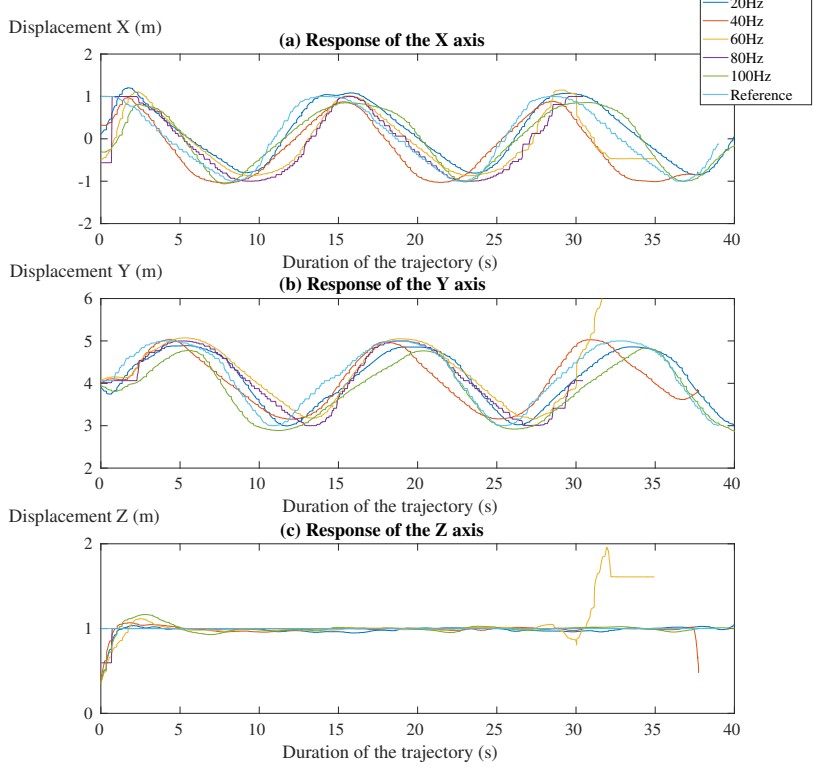

**Figure 8.** Response for a circular trajectory for various MPC rates and an MPC horizon of 80 steps.

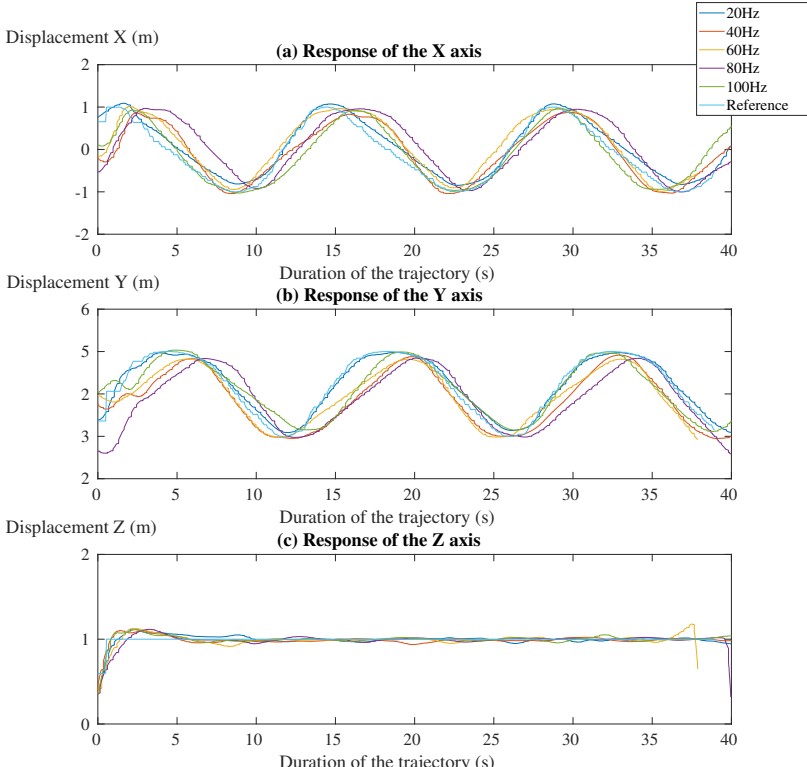

**Figure 9.** Response for a circular trajectory for various MPC rates and an MPC horizon of 100 steps.

The measurements for the circular trajectory, shown in Figure 4, were captured from multiple experiments run with every combination of different MPC horizons and rates. The MPC rates were set at 20 Hz, 40 Hz, 60 Hz, 80 Hz and 100 Hz while the MPC horizon steps were set to 20, 40, 60, 80 and 100. The responses of the circular trajectories are depicted in Figure 4, while the responses of each frame axis are depicted in Figures 5–9. From these figures, we are able to extract some valuable conclusions about our architecture and the behavior of the system regarding the MPC parameters that we chose to evaluate.

Figures 5–9 depict the responses. We can make an observation, shown more clearly in the figures regarding the response of the Z axis, that the overshoot decreases when we increase the MPC horizon. Since the MPC predicts the behavior of the system, it can follow the desired trajectory with the same velocity in a smoother manner by increasing the horizon.

The errors between the real and reference values of the response are depicted in Figure 10. In this figure, it is shown that the error after the CF takes off is always below the tolerance value of 0.4 m and, during the duration of the trajectory, is mainly under 0.25 m. This figure indicates the function of the controller, which ensures that the CF follows the desired trajectory.

In Figure 11, we can observe that by increasing the MPC rate, the round-trip time is drastically reduced. This was expected since the round-trip time is based partially on the execution time of the MPC. Additionally, we do not observe much deviation, nor any significant travel time delays, which can be explained by the fact that we used an edge computing framework that provides minimal latency and because the CF and the edge are operating in the same local network.

Regarding the CF microcontroller, the onboard processor is equipped with limited CPU power and memory, which could become a limitation if users want to implement new functionalities [27]. The 128KB RAM (the CF's onboard processor memory) is incapable of solving the optimization problem, and the execution crashes; thus, there is a need to utilize external resources (edge).

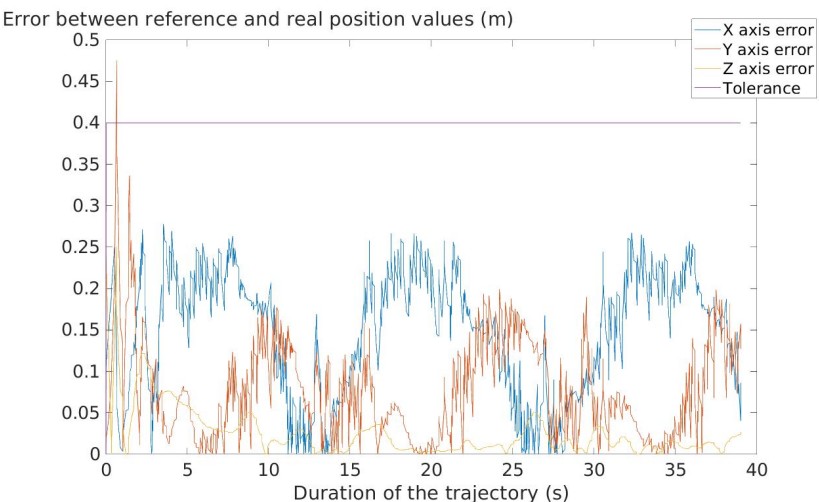

**Figure 10.** Errors between real and reference values of the X, Y and Z axes' responses.

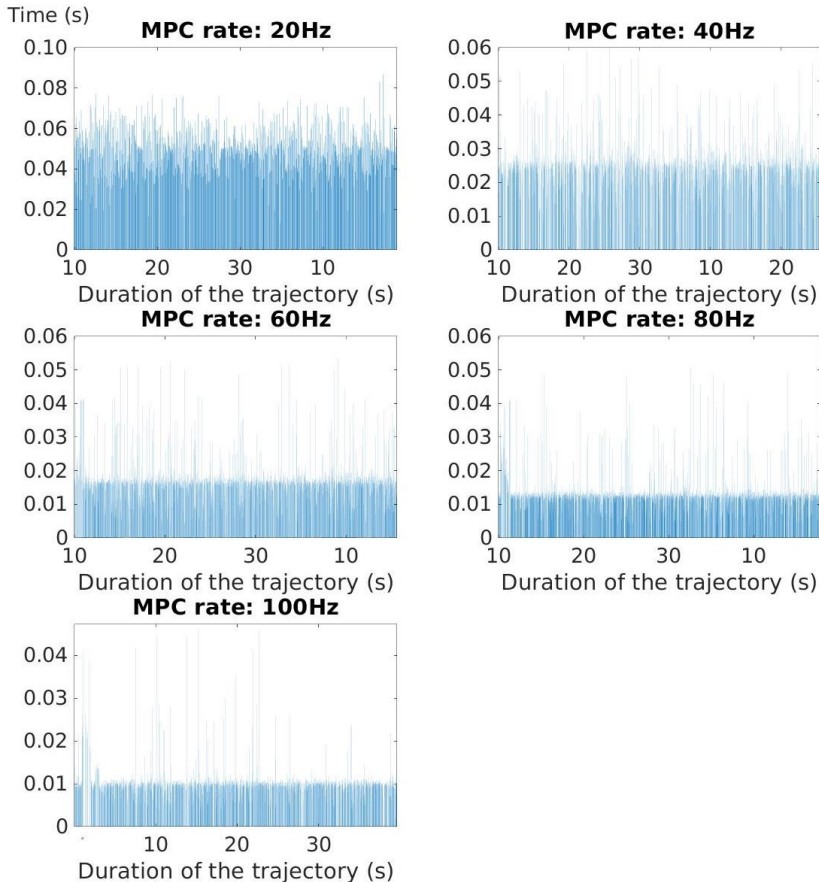

**Figure 11.** Round-trip time for a circular trajectory for various MPC rates and an MPC horizon of 100 steps.

The proposed edge architecture can be applied in multiple-use cases and for offloading various computational components while, in this case, we have used an aerial robot (the CF) that has no computational resources as a test-bed. To control the CF, we had to offload the overall high-level control architecture to the edge, a case that will be dominant in several scenarios in the future since the demand for more computationally heavy tasks is continuously increasing.

## 5. Conclusions and Future Work

In this article, we proposed a novel edge-based architecture-oriented MPC scheme with Kubernetes orchestration for controlling the trajectory of a UAV and, at the same time, held multiple experiments to evaluate the behavior of the proposed framework. As presented, we achieved control of the UAV to follow the desired trajectory, while the high-level motion controller ran on the edge. In the proposed architecture, Kubernetes was responsible for managing the deployed application on the edge and automating some procedures. Additionally, by using Kubernetes, we have the capability to mount and unmount resources on the fly, while Kubernetes takes control of the running applications and assigns them to the right work nodes automatically. These capabilities can be extremely handy in the future in cases in which it would be requested from the use case to have access to varying computational resources or if we run multiple applications on the edge for autonomous missions.

Two issues with our proposed system's architecture that need to be addressed are latency and safety concerns. These issues should be investigated further in the near future and are out of the scope of this article. An additional promising solution is 5G technology, but this might require specific configurations or have some network limitations. Safety should be investigated from two different perspectives. First, network safety should be examined and established. Second, it is essential to implement a high-level minimally computationally demanding backup controller that runs locally on the UAV's onboard processor. Hence, in the case of a network failure or a communication loss, the UAV will operate with the backup controller.

Moreover, our proposed architecture should be discussed in terms of capabilities and limitations. The motivation of our work was to increase the computational capabilities of resource-constrained robotic platforms, such as the CFs. By utilizing an edge machine, we were able to successfully reach our goal, but there are still some limitations. Since the proposed architecture is applied to almost real-time applications, the delays must remain minimal. The location and distance (in terms of network hops) of the edge machine from the robot should be taken into consideration to guarantee that the travel time delays will be bounded within a desired threshold. However, by offloading the application over the edge, the robots do not consume energy while executing the algorithms onboard; instead, they consume energy for communication purposes. For the CFs, the flying time remains relatively similar at about 5 to 7 min. Another limitation is the available edge resources. Edge providers do not offer "unlimited" resources, and, usually, the resources are quite costly. For our application, and with the available resources, we were able to tune the MPC to up to 100 steps of the prediction horizon. The system depends on the edge resources, and, if the applications require more resources than the available ones, the system can become overloaded.

Furthermore, edge architectures could be used for offloading many applications regarding UAV autonomous missions. Some interesting directions include multi-agent exploration and the multi-agent-based simultaneous localization and mapping (SLAM) applications in which multiple robots can explore an unknown environment, building maps on the edge while at the same time communicating and collaborating through the edge in real-time. The rapid development of communication (5G networks) and computing technologies (cloud, fog and edge computing) will provide more possibilities and solve current challenges, thereby helping the robotics field move towards fully autonomous systems.

**Author Contributions:** Conceptualization, A.S.S. and G.N.; Data curation, A.S.S.; Methodology, A.S.S. and B.L.; Software, A.S.S. and B.L.; Supervision, S.G.S. and G.N.; Writing—original draft, A.S.S.; Writing—review & editing, S.G.S., B.L. and G.N. All authors have read and agreed to the published version of the manuscript.

**Funding:** This research received no external funding.

**Institutional Review Board Statement:** Not applicable.

**Informed Consent Statement:** Not applicable.

**Data Availability Statement:** Not applicable.

**Conflicts of Interest:** The authors declare no conflict of interest.

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
