# Peer review of "An Edge-Based Architecture for Offloading Model Predictive Control for UAVs"

_robotics, doi:10.3390/robotics11040080_

Round 1

Reviewer 1 Report

This paper discusses a novel edge-based architecture oriented MPC scheme with Kubernetes orchestration for controlling the trajectory of a UAV.

- The topic discussed in this manuscript is interesting and fits the journal's scope.

- The novelty presented in the solution is adequate and the results seem promising. 

However:

- Information is missing on citation [1] "10 edge computing use case examples, 2022".

- "Offloading" should be a single word. The authors are advised to check it throughout the manuscript. 

- "Edge" is capitalised in some parts of the manuscript while not in some others. Is there a difference in the meaning?

- Some sentences are too long to read, please keep it short (e.g., line 29-33).

- Line 119. "Figure 1" lacks a space.

- What is the benefit of using a Kubernetes cluster for the offloading of the MPC controller, over using a plain VM infrastructure?

- Are there any constraints taken into consideration for the formulation of the MPC optimization problem?

- How do the two delays introduced, affect the trajectory planning for the UAV? 

- How does your work compare to on-board execution of the MPC controller? There's not really a comparison regarding the evaluation section.

- A similar interesting modern work on offloading the navigation procedures of a resource-constrained robotic device to the Edge (using also Control Theoretic techniques), is the following:

[1] Spatharakis, Dimitrios, et al. "A switching offloading mechanism for path planning and localization in robotic applications." 2020 International Conferences on Internet of Things (iThings) and IEEE Green Computing and Communications (GreenCom) and IEEE Cyber, Physical and Social Computing (CPSCom) and IEEE Smart Data (SmartData) and IEEE Congress on Cybermatics (Cybermatics). IEEE, 2020.

[2] Gräfe, Alexander, Joram Eickhoff, and Sebastian Trimpe. "Event-triggered and distributed model predictive control for guaranteed collision avoidance in UAV swarms." arXiv preprint arXiv:2206.11020 (2022).

Reviewer 2 Report

The paper is well structured and well written. However, the authors should improve its quality through considering these comments: 

Comment 1:

- The title is too long and should be modified

- The Figures 5, 6, 7, 8, 9 represent the responses for, should be “The Figures 5, 6, 7, 8 and 9 show the responses for”.

 Comment 2: After the Introduction the abbreviation should be defined before, they are used.

Example: MPC should be defined in the section “Introduction”.

 Comment 3: The introduction is too long. The authors should provide a separte paragraph of related works.

 Comment 4: What are the limitations of the proposed approach? What is the threshold of the control distance (steps)? What is the maximum duration of the trajectory?

 Comment 5: The authors proposed an offloading model for autonomous control of UAVs.   For the simulation scenarios, what were the quantities of offloaded data to the edge?

 Comment 6: How do the MPC rates and MPC horizon affect the offloading process and processing time?

Comment 7: The evaluation of the proposed system should be done by the comparisons of the obtained results to the recent state of the art.

Round 2

Reviewer 1 Report

The authors have addressed all my comments.

Reviewer 2 Report

The authors have done several efforts to improve the technical quality of the paper.